# Maternal Gdf3 is an obligatory cofactor in Nodal signaling for embryonic axis formation in zebrafish

**Brent W Bisgrove, Yi-Chu Su, H Joseph Yost***

Molecular Medicine Program, Eccles Institute of Human Genetics, University of Utah, Salt Lake City, United States

**Abstract** Zebrafish Gdf3 (Dvr1) is a member of the TGFβ superfamily of cell signaling ligands that includes *Xenopus* Vg1 and mammalian Gdf1/3. Surprisingly, engineered homozygous mutants in zebrafish have no apparent phenotype. Elimination of Gdf3 in oocytes of maternal-zygotic mutants results in embryonic lethality that can be fully rescued with *gdf3* RNA, demonstrating that Gdf3 is required only early in development, beyond which mutants are viable and fertile. *Gdf3* mutants are refractory to Nodal ligands and Nodal repressor Lefty1. Signaling driven by TGFβ ligand Activin and constitutively active receptors Alk4 and Alk2 remain intact in *gdf3* mutants, indicating that Gdf3 functions at the same pathway step as Nodal. Targeting *gdf3* and *ndr2* RNA to specific lineages indicates that exogenous *gdf3* is able to fully rescue mutants only when co-expressed with endogenous Nodal. Together, these findings demonstrate that Gdf3 is an essential cofactor of Nodal signaling during establishment of the embryonic axis.

DOI: https://doi.org/10.7554/eLife.28534.001

## Introduction

Nodal, a member of the TGFβ superfamily of cell-cell signaling ligands, is essential in the establishment of vertebrate axis patterning, both the primary embryonic axis during gastrulation and the orthogonal left-right (LR) asymmetries in heart, brain and gut development. *vg1*, another member of the TGFβ family, was prototypically described as the first-known regionally localized RNA in a vertebrate oocyte, localized to the vegetal pole of the egg, and subsequently in early embryos (*Rebagliati et al., 1985*). Humans and other mammals have two *vg1* orthologues, *GDF1* and *GDF3*, and zebrafish have a single orthologue, *gdf3*, previously known as *dvr1*.

Although *gdf3* (*vg1*) has been a target of considerable study for many years, genetic mutations for functional analysis have not been reported in zebrafish. Studies in *Xenopus* have suggested that maternal Vg1 is required for early zygotic expression of anterior mesendodermal genes (*Birsoy et al., 2006*). Vg1 has been implicated in early Left-Right patterning, as overexpression of *Xenopus* Vg1 fusion proteins or mouse Gdf1 fusion proteins in specific early cell lineages can fully invert the LR axis (*Hyatt et al., 1996*; *Hyatt and Yost, 1998*; *Wall et al., 2000*). Vg1 is the only known member of the TGFβ family with this early LR patterning capability. At later stages of development, overexpression or grafts into lateral plate mesoderm that activate TGFβ family member Xenopus Nodal (*Xnr1*) in right lateral plate can also invert asymmetry (*Ohi and Wright, 2007*). In mice, homozygous knockout mutations have altered asymmetry (*Rankin et al., 2000*). In humans, genetic variants in *GDF1* have been implicated in complex congenital heart defects (*Karkera et al., 2007*; *Zhang et al., 2015*), likely to be effects of upstream altered LR patterning. Similar to *vg1* in *Xenopus*, *gdf3* RNA and protein are abundantly stored in the oocyte and early embryo before zygotic gene activation in zebrafish (*Helde and Grunwald, 1993*; *Peterson et al., 2013*). In zebrafish, Kupffer's vesicle (KV), the ciliated organ of asymmetry (*Essner et al., 2005*; *Essner et al.,*

*For correspondence:
jyost@genetics.utah.edu

**Competing interests:** The authors declare that no competing interests exist.

**eLife digest** All vertebrates – animals with backbones like fish and humans – have body plans with three clear axes: head-to-tail, back-to-front and left-to-right. Animals lay down these plans as embryos, when signaling molecules bind to receptors on the surface of their cells.

These signaling molecules include related proteins called "Nodal" and "Growth and Differentiation Factors". However, there has been much debate in the field of developmental biology about whether these proteins work together or independently during the early development of vertebrates.

Zebrafish are often used to study animal development, and Bisgrove et al. decided to test whether these fish need a Growth and Differentiation Factor known as Gdf3 by deleting it using genome editing. It turns out that zebrafish can survive and develop as normal without the gene for Gdf3, just as long as their mothers still had a working copy of the gene. Yet, when the offspring of mutant females did not inherit the instructions to make Gdf3 from their mothers, they died within a couple of days. This was true even if the offspring inherited a working copy of the gene from their fathers. Bisgrove et al. then went on to show that embryos from a mutant mother could be saved with an injection of short-lived RNA molecules that include the instructions to make some Gdf3 proteins. The injected mutant embryos could live to adulthood. This shows that Gdf3 is only needed during the embryo's early development.

Further experiments suggested that Gdf3 does cannot activate its receptors on its own. Instead, it is likely that Gdf3 interacts with Nodal to form a two-protein complex that activates the receptors. Two other groups of researchers have independently reported similar findings.

Mutations affecting proteins very similar to Gdf3 have been found in people with congenital heart defects. By revealing the interaction between Gdf3 and Nodal, these new findings could help scientists to understand the genetic causes of this condition in more detail. Further studies using the mutant zebrafish could also be used to explore the causes of other developmental diseases.

DOI: https://doi.org/10.7554/eLife.28534.002

*2002*), contains motile cilia that generate asymmetric fluid flow and LR patterning information that is transmitted to lateral plate mesoderm (LPM), which then conveys LR patterning information to the brain, heart and gut primordia (*Dasgupta and Amack, 2016*). Zebrafish *gdf3* is expressed in tissues implicated in LR patterning including cells adjacent to the KV, in the LPM and in heart primordia. Morpholino knockdown of *gdf3* results in normal KV structure, KV cilia length and motility, and normal asymmetric KV fluid flow, but disruption of downstream LR patterning in the LPM. Thus, zebrafish Gdf3 was proposed to transmit LR information generated by KV cilia flow to LPM (*Peterson et al., 2013*), and the reasonable expectation was that mutants of *gdf3* in zebrafish would have LR patterning defects.

We engineered mutants in zebrafish to test the roles of *gdf3*. Surprisingly, homozygous zygotic mutants develop normally and do not show alterations in LR patterning. On the other hand, zebrafish mutants in which the maternal stores of Gdf3 are eliminated have a much more dramatic embryonic lethal phenotype, which we analyzed for a wide range of tissue specification and embryonic patterning pathways. Results from a series of epistasis experiments and cell-lineage targeted rescues of *gdf3* mutants indicate that Gdf3 functions as an obligate co-factor of Nodal, and that Nodal functions as an obligate cofactor of Gdf3. These results indicate that neither Nodal nor Gdf3 can function without the other in fundamental patterning of the vertebrate embryonic axis.

## Results

### Zygotic *gdf3* mutants are viable and fertile

We generated mutant alleles of *gdf3* by using TALENs designed to target genomic sequences encoding the first few amino acids of the pro-domain of Gdf3 (*Figure 1A*; *Figure 1—figure supplement 1*). High resolution melt analysis (HRMA) (*Parant et al., 2009*) was used to screen for mutations in genomic DNA from embryos of G0 founders and later, from fin clips from adults derived from those founders (*Figure 1B*). Among several mutants identified, three lines were selected and

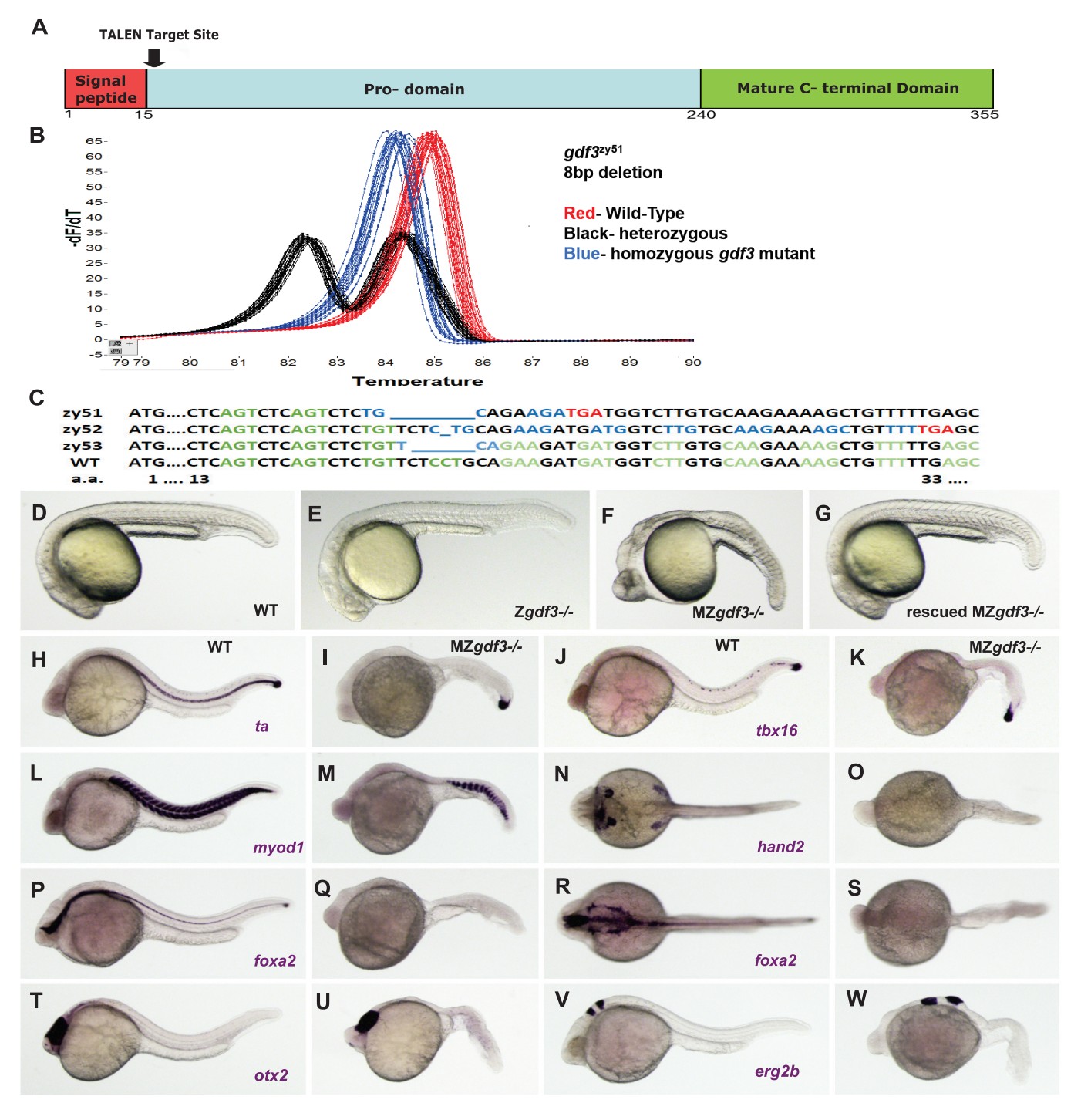

**Figure 1.** Zygotic *gdf3* mutants are viable and maternal zygotic mutants have phenotypes indicative of loss of Nodal signaling. (A–C) *gdf3* mutants were generated using TALEN-mediated mutagenesis. (A) *Gdf3*, a TGFβ family member, comprises a signal peptide, pro-domain and mature TGFβ domain. TALENs were designed to target genomic sequences located near the amino end of the pro-domain. (B) Mutants were identified by high-resolution melt analysis (HRMA). (C) Three mutant alleles, *gdf3^zy51^*, *gdf3^zy52^*, *gdf3^z53^*, had 8, 1 and 6 bp deletions respectively. (D–G) Morphological phenotypes of *gdf3* mutants at 24 hpf. (D) Wild-type (WT) and (E) zygotic (Z) *gdf3* mutants were phenotypically indistinguishable. (F) Maternal-zygotic (MZ) *gdf3* mutants lacked notochord, pharyngeal endoderm and had reduced anterior neural tissues. (G) MZ*gdf3* mutants were completely rescued by injection of 100 pg *gdf3* RNA at the one-cell stage. (H–W) WISH analysis of gene expression in WT (columns H-T and J-V) and MZ*gdf3* (columns I-U and K-W) mutants at 24 hpf. (H, I) *ta* (*ntl*) expression in the notochord was absent from MZ*gdf3* although tailbud expression was maintained. (J, K) *tbx16* (*spt*) expression in spinal cord neurons was absent in MZ*gdf3* while tailbud expression is unaffected. (L, M) Trunk and tail somites expressing *myod1*

*Figure 1 continued on next page*

*Figure 1 continued*

were reduced in number and altered in shape in MZ*gdf3*. (**N, O**) Expression of *hand2* in the heart, pharyngeal arch mesoderm and pectoral fin buds is absent in MZ*gdf3*. (**P–S**) Ventral neural tissues and pharyngeal endoderm expressing *foxa2* (*axial*) were absent in MZ*gdf3*. Patterns of expression of *otx2* in the forebrain and midbrain (**T, U**), and *erg2b* (*krox20*) in hindbrain rhobomeres 3 and 5 (**V, W**) were altered in MZ*gdf3* mutants. All embryos (**D–W**) are shown in lateral view with rostral to the left except N, O, R, S which are dorsal views in the same orientation. Each panel is a representative image from at least 15 embryos.

DOI: https://doi.org/10.7554/eLife.28534.003

The following figure supplements are available for figure 1:

**Figure supplement 1.** Map of the genomic sequences around the first coding exon of *gdf3*.

DOI: https://doi.org/10.7554/eLife.28534.004

**Figure supplement 2.** The Left-Right patterning marker *spaw* is expressed in zygotic *gdf3* mutants.

DOI: https://doi.org/10.7554/eLife.28534.005

**Figure supplement 3.** High magnification views of wild-type and MZ*gdf3* mutant embryos.

DOI: https://doi.org/10.7554/eLife.28534.006

**Figure supplement 4.** Phenotypic classes of MZ*gdf3* rescued with *gdf3* RNA.

DOI: https://doi.org/10.7554/eLife.28534.007

**Figure supplement 5.** (A, B) RT-PCR analysis of cDNA prepared from 1000 cell-, 90% epiboly-, and 18 somite-stage embryo RNA, respectively, from: (Lanes 1–3) WT embryos, (Lanes 4–6) embryos derived from a female MZ*gdf3*$^{zy51}$ X WT male, and (Lanes 7–9) MZ*gdf3*$^{zy51}$ embryos.

DOI: https://doi.org/10.7554/eLife.28534.008

propagated. The results presented here utilize *gdf3*$^{zy51}$ and *gdf3*$^{zy52}$ alleles; both mutations result in reading frame shifts which encode missense amino acids and early termination codons, suggesting they may be functional nulls (*Figure 1C*). The phenotypes are indistinguishable between *gdf3* mutants carrying either the *zy51* or *zy52* allele and these mutants are used and described interchangeably. A third mutant line, *gdf3*$^{zy53}$, has a 6 bp in-frame deletion that removes two amino acids but leaves the predicted protein otherwise unchanged and had no observable phenotype.

Surprisingly, homozygous recessive zygotic mutants generated by in-crosses of heterozygous carriers of either *gdf3*$^{zy51}$ or *gdf3*$^{zy52}$ had no morphological differences from wild-type (WT) embryos (N > 100) (*Figure 1D,E*). Homozygous mutants survive to adulthood at normal wild-type rates. In contrast to previous results with morpholinos in zebrafish, zygotic mutants of *gdf3* had normal LR patterning as assessed by scoring heart looping in clutches of embryos derived from heterozygous *gdf3* in-crosses at 48 hr post-fertilization (hpf). As in WT, clutches of Z*gdf3* embryos showed about a 2% heart reversal rate (Z*gdf3*$^{zy51}$ 64/65 normal heart looping, 2 clutches scored; Z*gdf3*$^{zy52}$ 154/157 normal heart looping, 3 clutches scored; WT 164/168 normal heart looping, 3 clutches scored). Consistent with the heart looping data, an examination of the expression of the early Left-Right marker gene *spaw* at the 18–20 somite-stage in embryos derived from WT parents or from an incross of *gdf3*$^{zy51}$ heterozygotes showed clutches of WT and Z*gdf3* with similar percentages of normal left-sided gene expression (93% in WT clutches; 90% in Z*gdf3* clutches) (*Figure 1—figure supplement 2*). One quarter of the adults raised from in-crosses of *gdf3* heterozygotes were homozygous for the mutant alleles as assayed by HRMA, indicating no embryonic or adult lethality of Z*gdf3*.

## The maternal genome, not the paternal genome, dictates the phenotype of embryos

Homozygous zygotic *gdf3* mutant adults are fertile. In-crosses of homozygous mutants yielded Maternal Zygotic (MZ*gdf3*) embryos that exhibit a pronounced phenotype at 24 hpf, with 100% penetrance (N > 100) (*Figure 1F*). At 24–26 hpf, MZ*gdf3* embryos lack a notochord and spinal cord and structures associated with lateral plate mesoderm including the heart, and have greatly reduced anterior neural development reflected in the narrowed neural tube, un-folded brain and reduced tissue between the eyes resulting in cyclopia (*Figure 1—figure supplement 3*).

To confirm that the MZ*gdf3* phenotype was due to a loss of wild-type *gdf3* gene product, we asked whether MZ*gdf3* embryos could be rescued by *gdf3* mRNA injections (*Figure 1G*). To assess the dose dependence of rescue, we devised a phenotype classification system (*Figure 1—figure supplement 4*) somewhat analogous to the DAI (dorso-anterior indices) in *Xenopus* (*Kao and Elinson, 1988*) that has been useful for studies of axis formation. Injection of 100 pg *gdf3* mRNA fully rescued over 90% of the MZ*gdf3* embryos, and these fully rescued embryos could survive until

adulthood and survivors were fertile. For example, of 22 rescued MZ*gdf3* embryos selected for an inflated swim bladder at 6 days post-fertilization and raised to adulthood, 14 survived. From those, 5 mating pairs produced clutches of embryos with the MZ*gdf3* phenotype at 24 hpf, demonstrating fertility of both males and females that were derived from Gdf3 null embryos that received only a pulse of WT *gdf3* mRNA shortly after fertilization.

To assess whether both maternal and paternal contributions to the zygote were necessary for the MZ*gdf3* phenotype, we performed reciprocal pairwise crosses of male and female MZ*gdf3* homozygotes to either Z*gdf3* heterozygotes or to wild-type fish and found that the mutant phenotype of the derived embryos was solely dependent on a *gdf3*$^{-/-}$ maternal genotype; the paternal genome did not alter the phenotype. Embryos derived from *gdf3*$^{-/-}$ males showed no mutant phenotype when derived from a heterozygous female with a wildtype allele of *gdf3* (n = 4 mating pairs).

Strikingly, paternal contribution of a wild-type *gdf3* allele could not rescue embryos derived from MZ*gdf3*- females (n = 7 mating pairs with *gdf3*+/− male; n = 3 mating pairs with WT +/+ males). In other words, *gdf3*+/− heterozygotes that were derived from *gdf3* null eggs and wildtype sperm displayed the MZ*gdf3* phenotype. To assess whether the paternal genome was transcriptionally activated following zygotic gene activation (ZGA) in embryos from a cross of MZ*gf3* female by WT male cross, we carried out RT-PCR to compare cDNA samples prepared from pre-ZGA 1000 cell-stage and post-ZGA 90% epiboly, and 18 somite-stage embryos, and compared those to samples at the same time points derived from embryos from a MZ*gdf3* in-cross and a WT cross (*Figure 1—figure supplement 5*). There was a substantial decrease in the amount of PCR product from the 1000 cell stage to the 18-somite stage in each genotype, but we were unable to detect any substantial differences among samples prepared from the same time points across the three genotypes indicating that the mutant maternal RNA from MZ*gdf3* embryos was as stable as that from WT embryos (*Figure 1—figure supplement 5A,B*). To resolve whether the paternal genome was activated in a *gdf3*-maternal background, we designed primers specific to the WT allele of *gdf3* and carried out RT-PCR on the same series of stages and genotypes (*Figure 1—figure supplement 5C,D*). WT *gdf3* transcript was detected in all stages of WT embryos (lanes C1-3) and was absent from cDNA derived from homozygous MZ*gdf3* embryos (serving to indicate that the WT primers do not amplify the maternal or paternal mutant allele; lanes C7-9). WT transcript was also undetectable in pre-ZGA stage embryos (1000 cell stage, lane 4) derived from a cross of a MZ*gdf3* mutant female by WT male, but was present in cDNA derived from post-ZGA stages (lanes C4, 5). The fact that there is less transcript detected at 90% epiboly and 18 somite stages of embryos derived from a MZ mutant female versus a WT female may reflect perdurance of maternal WT transcript from the WT female, that there is significantly less mesoderm in MZ mutant embryos in which to express gdf3, or that the paternal genome is expressed to a lesser extent in embryos lacking WT maternal transcript (or a combination thereof). Although the paternal WT genome is clearly activated in MZ*gdf3* embryos, this post-ZGA expression of paternal WT *gdf3* does not rescue the MZ phenotype.

## Gdf3 is required for mesoderm, endoderm and neural patterning

To further characterize the phenotypes of MZ*gdf3* embryos, we carried out whole mount in situ hybridization (WISH) in wild-type and MZ*gdf3* embryos using several tissue-specific gene expression markers at 24hpf. The T-box transcription factor *ta* (*ntl*) is expressed in the notochord, a derivative of the axial mesoderm, and in the tailbud mesenchyme (*Figure 1H*). In MZ*gdf3* embryos, the notochord was absent as evidenced by the lack of *ta* expression in the midline; expression in the tailbud remains (*Figure 1I*). Expression of another T-box transcription factor *tbx16* (*spt*), expressed in paraxial and ventral mesoderm, was maintained in the tailbud of MZ*gdf3* embryos but *tbx16* expression in a subset of spinal cord neurons in WT embryos was not detected in MZ*gdf3* mutants (*Figure 1J, K*). Similarly, the ventral mesoderm marker *eve1* was expressed in the tailbud of both WT and MZ*gdf3* embryos (not shown). The myogenic transcription factor *myod1* was expressed in chevron-shaped somites extending the length of the trunk and tail of embryos. In similar stage MZ*gdf3* embryos, *myod1* expression occurred in fewer, smaller, u-shaped somites restricted to the caudal trunk and tail (*Figure 1L,M*). In WT embryos *hand2* was expressed in the heart tube, a derivative of the anterior lateral plate mesoderm, in the pharyngeal mesoderm of the branchial arches and in the pectoral fin bud. All of these tissue-specific expression domains were absent in MZ*gdf3* (*Figure 1N, O*). The forkhead box transcription factor *foxa2* (*axial*) was expressed in the midline in the ventral brain, the spinal floorplate and the chorda-neural hinge as well as laterally in the pharyngeal

endoderm of the branchial arches. All of these expression domains were absent in MZ*gdf3* mutant embryos (*Figure 1P–S*). The rostral-most regions of the neural tube were relatively unaffected in MZ*gdf3* embryos. Expression domains of *otx2* in the forebrain and midbrain, and *egr2b* (*krox20*) in hindbrain rhombomeres 3 and 5 were present though somewhat misproportioned and misshapen in MZ*gdf3* mutant embryos versus WT embryos (*Figure 1T–W*). Together, these results indicate that maternally provided Gdf3 is essential in the patterning of mesoderm, endoderm and in some aspects of neural development.

## Maternal zygotic *gdf3* mutants have altered downstream pathways indicative of loss of Nodal signaling and are rescued by *gdf3* RNA injection

To investigate roles of Gdf3 during germ layer formation and early tissue patterning, we assessed the consequences of loss of Gdf3 function, gain of function and functional rescue on the patterns of tissue-specific gene expression markers by injecting WT and MZ*gdf3* embryos with *gdf3* RNA at the 1–2 cell stage. Injected or un-injected embryos were analyzed by WISH with tissue-specific markers at 90% epiboly (*Figure 2*).

The MZ*gdf3* phenotypes described above are strikingly similar to phenotypes of mutant embryos lacking maternal Nodal signaling components such as in double homozygous mutants for *ndr1;ndr2* (*sqt;cyc*), MZ*tdgf1* (MZ*oep*) and MZ*foxh1* (*Dougan et al., 2003*; *Gritsman et al., 1999*; *Slagle et al., 2011*), or from overexpression of the Nodal antagonists *lft1* or *lft2* (*Bisgrove et al., 1999*; *Thisse and Thisse, 1999*). In all cases, these mutants or treated embryos exhibit a loss of notochord and spinal cord and reduction and disorganization in trunk and tail somites as well as a severe reduction in anterior and neural structures that manifest as a loss of brain patterning and synophthalmia as observed in MZ*gdf3* (see *Figure 1—figure supplement 3*). Thus, we examined expression of Nodal signaling pathway genes *ndr2*, *lft1* and *lft2* (*Figure 2A–L*). The Nodal family member *ndr2* was expressed in anterior and posterior axial mesoderm and anterior axial neurectoderm (*Figure 2A*). This WT expression pattern was unchanged in embryos injected with *gdf3* RNA (*Figure 2B*). In contrast, MZ*gdf3* embryos were void of any detectable *ndr2* expression, but the normal expression pattern could be completely restored by injection of *gdf3* RNA (*Figure 2C,D*).

Since a nodal-deficiency phenotype can be obtained by overexpression of Nodal antagonists *lft1* or *lft2* (*Bisgrove et al., 1999*), we asked whether the MZ*gdf3* phenotype was due to hyperexpression of *lft1* or *lft2*. The normal expression patterns of *lft1* in marginal mesendoderm, the axial chordamesoderm, prechordal plate mesoderm and anterior ventral neurectoderm) and *lft2* in posterior axial neurectoderm and prechordal plate mesoderm (*Figure 2E,I*) were absent in MZ*gdf3* embryos (*Figure 2G,K*). Neither gene expression pattern was altered in WT embryos injected with *gdf3* RNA (*Figure 2F,J*), and *gdf3* RNA injection fully rescued expression of both of these Nodal response genes in MZ*gdf3* (*Figure 2H,L*). These results, and the observation that *lft1* and *lft2* expression are absent from MZ*gdf3* embryos at shield stage during mesendoderm formation (*Figure 2—figure supplement 1*), indicate that the MZ*gdf3*-deficient phenotype was not due to hyperexpression of endogenous *lft1* and *lft2*.

We next examined expression of mesodermal genes that are not Nodal (TGFβ) family members (*Figure 2M–X*). The transcription factors *ta* and *gsc* are direct targets of Nodal signaling and overlap with the expression of *ndr2* in the midline. (*Chen and Schier, 2001*; *Thisse et al., 1994*) In WT embryos, *ta* was expressed in posterior axial chordamesoderm that gives rise to the notochord as well as in the marginal mesendoderm, while *gsc* was expressed in the anterior prechordal plate mesoderm, which gives rise to the polster, and in anterior ventral neurectoderm (*Figure 2M*). *gsc* and *ta* expression patterns were unchanged in embryos injected with *gdf3* RNA (*Figure 2N*). In MZ*gdf3* embryos midline *gsc* and *ta* expression domains were absent while *ta* expression in the marginal mesendoderm was unaffected (*Figure 2O*). Injection of *gdf3* RNA fully rescued the axial expression domains of both *ta* and *gsc* (*Figure 2P*). The t-box transcription factor *tbx16* was expressed in a broad domain of paraxial mesendoderm at the margin as well as in the midline in the prechordal plate mesoderm (Fig. Q). In MZ*gdf3* mutant embryos axial *tbx16* expression in the prechordal plate was absent and the width of the paraxial expression domain was reduced (*Figure 2S*). Injection of *gdf3* RNA into WT embryos did not alter normal *tbx16* domains but was capable of fully rescuing expression in the prechordal plate and expanding the paraxial expression domain to a normal width (*Figure 2T*). The homeobox transcription factor *eve1* was expressed in a broad domain at

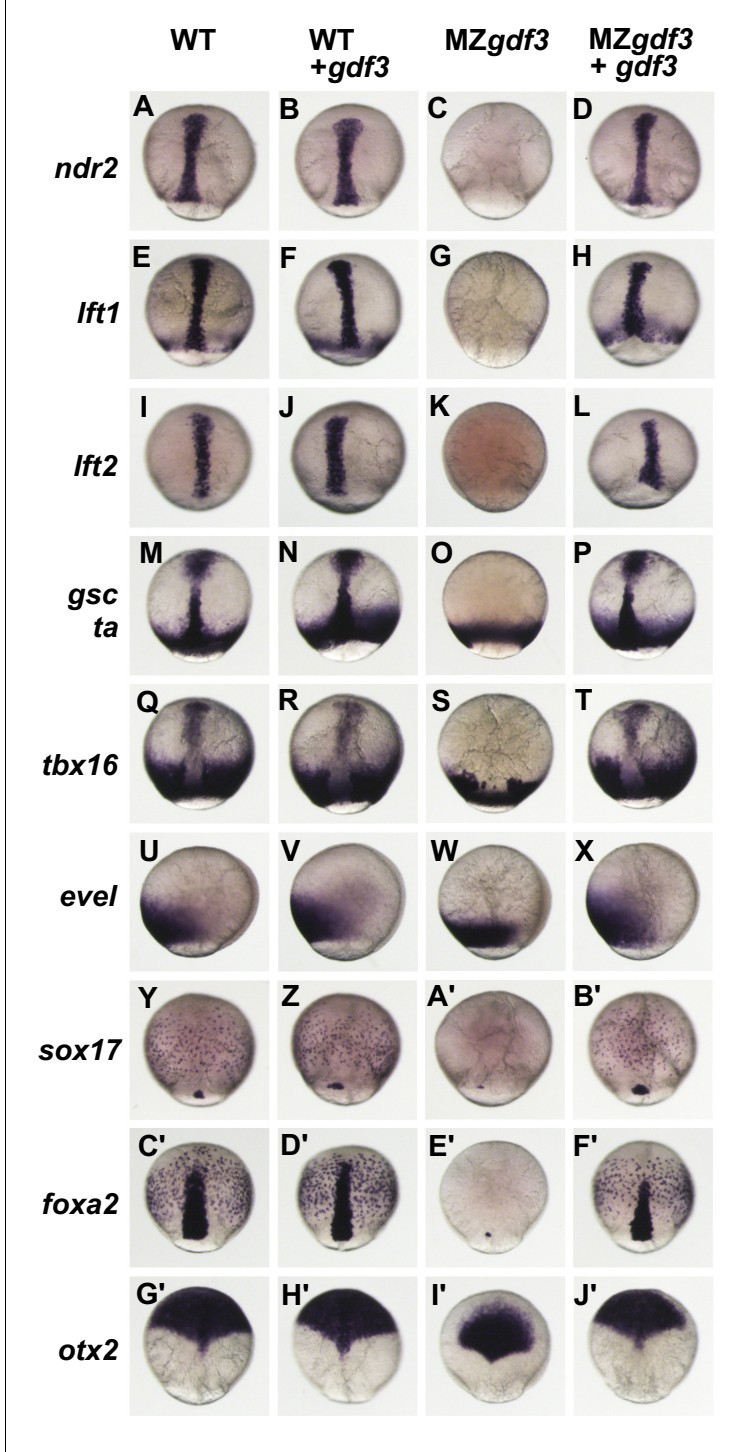

**Figure 2.** Gdf3 is required for mesoderm, endoderm and neural patterning. (**A–J'**) WISH analysis of gene expression in embryos at 90% epiboly. Columns from left to right show WT embryos, WT embryos injected with 100 pg of *gdf3* RNA, MZ*gdf3* mutants and MZ*gdf3* injected with *gdf3* RNA. Each panel is a representative image of at least 15 embryos examined. (**A–L**) Midline and margin expression of Nodal signaling pathway genes *ndr2* (**A–D**) and Lefty family members *lft1* (**E–H**) and *lft2* (**I–L**) were absent in MZ*gdf3* mutants and restored by *gdf3* mRNA injection. (**M–X**) Analysis of early mesoderm transcription factor genes. (**M–T**). Expression domains of *gsc*, *ta* (*ntl*) and *tbx16* (*spt*) were absent from the midline of MZ*gdf3* mutants, but restored by *gdf3* RNA injection. (**Q–X**) Lateral and ventral mesendoderm expression domains of *tbx16* (**Q–T**), and *eve1* (**U–X**), which were reduced in width in MZ*gdf3*, were restored to wild-type levels by *gdf3* RNA. (**Y–F'**) Endoderm expression domains of

*Figure 2 continued on next page*

*Figure 2 continued*

transcription factors *sox17* (Y–B') and *foxa2* (C'–F') were absent in MZ*gdf3*, and restored by *gdf3* RNA, as was expression of *foxa2* in midline neural tissues. (G'–J'). *otx2* expression in the anterior neural plate is reduced in MZ*gdf3* but rescued to its normal extent by *gdf3* RNA. (Second column from left) Strikingly, although injection of *gdf3* RNA was capable of rescuing mesoderm, endoderm and neural tissues in MZ*gdf3* mutants it had no effect on gene expression in WT embryos.

DOI: https://doi.org/10.7554/eLife.28534.009

The following figure supplement is available for figure 2:

**Figure supplement 1.** Nodal antagonists *lft1* and *lft2* are not expressed in MZ*gdf3*.

DOI: https://doi.org/10.7554/eLife.28534.010

---

the ventral margin of WT embryos and like other genes with margin expression was unaffected by ectopic expression of *gdf3* (**Figure 2U,V**). In MZ*gdf3* embryos the width of this expression domain was reduced, but was fully restored to normal width by the injection of *gdf3* at the 1–2 cell stage (**Figure 2W,X**).

The roles of Gdf3 in non-mesodermal germ layers were examined using probes to the transcription factors *sox17*, *foxa2* and *otx2* (**Figure 2Y–J'**). *Sox17* is normally expressed in presumptive endoderm cells scattered throughout the blastoderm and in dorsal forerunners, a group of non-involuting cells at the dorsal margin (**Figure 2Y**) that form the KV. Neither of these cell types is affected by the injection of *gdf3* into WT embryos (**Figure 2A'**). In MZ*gdf3*, no *sox17*-expressing cells were detected in the blastoderm and in most embryos (14 of 16) no expression was detected in cells at the dorsal margin. In the remaining few cases, one or two *sox17* positive cells were detected at the dorsal margin (**Figure 2A'**). *sox17* expression in the presumptive endoderm and dorsal forerunner cells of MZ*gdf3* was restored by injection of *gdf3* (**Figure 2B'**). Like *sox17*, *foxa2* was also expressed in presumptive endoderm, and additionally in axial mesendoderm and neurectoderm (**Figure 2C'**). None of these WT expression domains was affected by overexpression of *gdf3* (**Figure 2D'**). In MZ*gdf3* embryos *foxa2* expression in the endoderm was absent and in the majority of embryos, expression in the axis was absent. In a few embryos (3 of 15) one or two *foxa2*-expressing cells were present at the dorsal margin (**Figure 2E'**). *foxa2* expression in MZ*gdf3* embryos was rescued to a normal pattern by injected *gdf3* RNA (**Figure 2F'**). The anterior neural plate, marked by triangle-shaped expression domain of *otx2* that extends from the midpoint of the dorsal axis over the animal pole, was unchanged in WT embryos by overexpression of *gdf3* (**Figure 2G', H'**). In MZ*gdf3* embryos a triangular *otx2* expression domain was present but greatly reduced in size, failing to extend to the animal pole (**Figure 2I'**). Like all of the marker genes examined here, *otx2* expression was fully restored to a normal pattern in MZ*gdf3* embryos by expression of exogenous *gdf3* RNA (**Figure 2J'**). Together, these results indicate that Nodal signaling response pathways are dependent on Gdf3 function.

## MZ*gdf3* mutants are refractory to Nodal ligands and inhibitors

To investigate the position of Gdf3 in the Nodal signaling pathway, we expressed other members of that pathway in WT and MZ*gdf3* by injecting RNAs encoding those proteins at the 1–2 cell stage and assaying the resultant phenotypes at shield stage and at 24 hpf (**Figure 3**). In WT shield-stage embryos *gsc* was expressed in a dorsal crescent of cells that occupied about 15% of the circumference of the germ ring, while *ta* was expressed throughout the germ ring (**Figure 3A,B**). Shield-stage embryos of MZ*gdf3* lacked *gsc* expression and *ta* expression was reduced dorsally (**Figure 3D,E**) compared to WT embryos. Injection of *gdf3* in WT embryos had no effect on the expression of *gsc* or *ta* and did not alter morphological phenotype at 24 hpf (**Figure 3G-I**). Injection of *gdf3* into MZ*gdf3* embryos rescued dorsal expression of *gsc* and *ta* (**Figure 3J,K**) and morphology at 24 hpf (**Figure 3L**).

Exogenous expression of RNAs encoding Nodal family members *ndr1* and *ndr2* in WT embryos caused ectopic expression of both *gsc* and *ta* in the margin germ ring and blastoderm at shield stage and resulted in dorsalized phenotypes at 24 hpf (**Figure 3M–O,S–U**). This serves as a positive control for the power of these ligands to activate Nodal signaling. In contrast, injection of *ndr1* or *ndr2* RNA into MZ*gdf3* embryos did not rescue *gsc* or *ta* expression in the dorsal germ ring of shield stage embryos, did not rescue axial tissues in embryos at 24 hpf, and did not induce a dorsalized

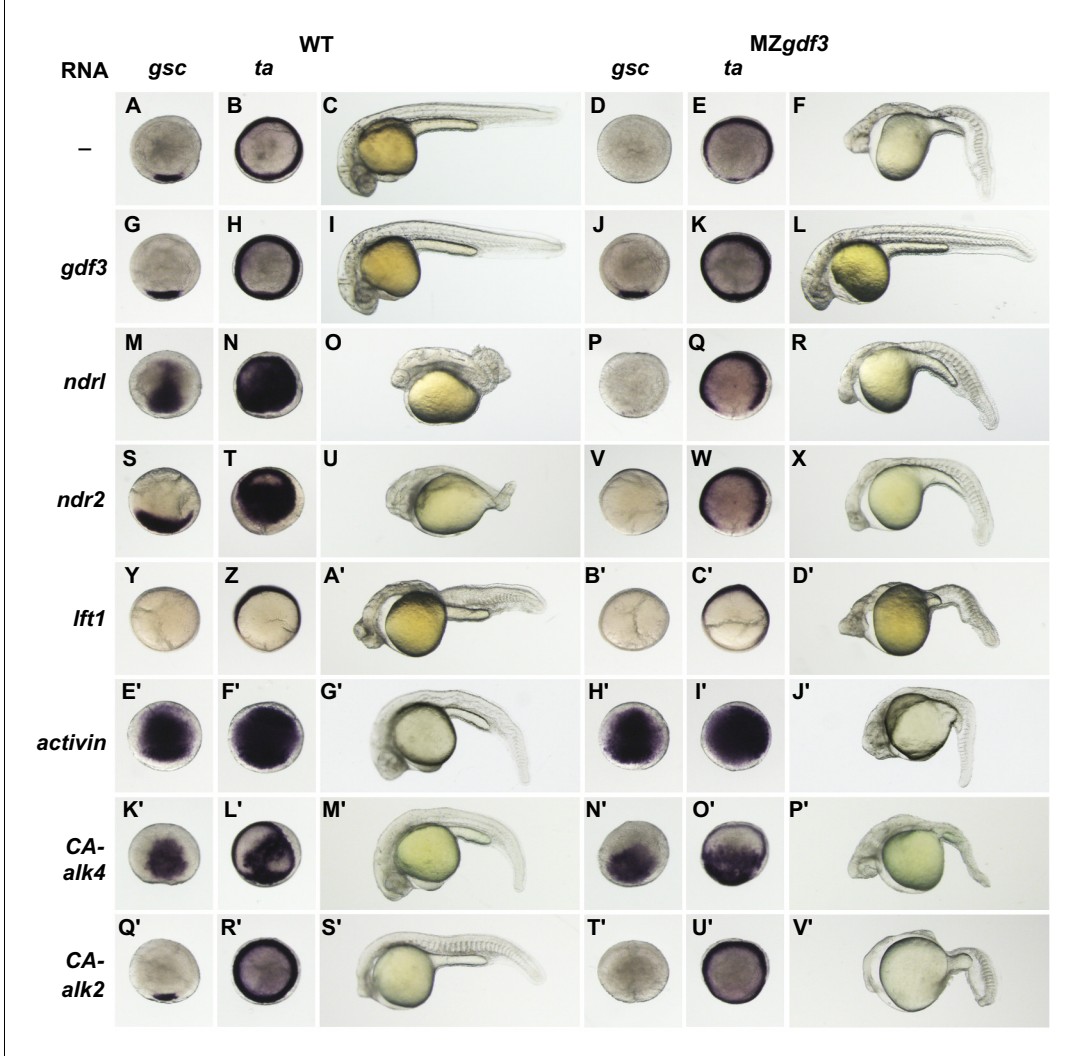

**Figure 3.** MZ*gdf3* mutants are refractory to Nodal ligands and inhibitors, yet respond to other TGFβs and have an intact downstream response pathway. (**A–V'**) WT and MZ*gdf3* embryos injected at 1–2 cell stage with RNAs encoding *gdf3*, other TGFβ pathway members, inhibitors of Nodal, and downstream constitutively active (CA) *Alk4* and *Alk2* receptors. Panels show WISH expression patterns of *gsc* and *ta* in animal pole views of shield stage embryos (dorsal toward bottom) and light micrographs of morphology at 24 hpf. Each panel is a representative image of at least 15 embryos examined. (**A–C**) Un-injected WT and (**D–F**) MZ*gdf3* embryos. (**G–I**) Wild-type embryos injected with 100 pg *gdf3* RNA were unaffected (**J–L**). Injected *gdf3* RNA completely rescued *gsc* and *ta* expression patterns and morphology at 24 hpf in MZ*gdf3* embryos. Injected Nodal RNAs *ndr1* (10 pg) (**M–O**) and *ndr2* (10 pg) (**S–U**) caused ectopic expression of *gsc* and *ta* at shield, and dorsalized embryos at 24 hpf. In contrast, (**P–R**) *ndr1* RNA or (**V–X**) *ndr2* RNA injection into MZ*gdf3* embryos failed to rescue dorsal *gsc* or *ta* expression at shield stage and did not alter mutant morphology at 24 hpf. (**Y–A'**) Injection of 25 pg RNA encoding the Nodal inhibitor *lft1* into WT embryos resulted in a loss of dorsal *gsc* and *ta* expression at shield stage similar to that seen in MZ*gdf3* embryos, and with phenotypes resembling MZ*gdf3* at 24 hpf. (**B'–D'**) In MZ*gdf3* injected with *lft1* the *gsc* and *ta* expression patterns and 24 hpf mutant phenotype were similar to un-injected MZ*gdf3*. Phenotypes of *lft1*-injected embryos appear more severe than MZ*gdf3* because these embryos are slightly older. Injection of 2.5 pg RNA encoding *Xenopus* Activin, a general TGFβ pathway activator, induced ectopic expression of *gsc* and *ta* in both (**E'–G'**) WT and (**H'–J'**) MZ*gdf3* embryos, and dorsalization of embryos that survived to 24 hpf. (**K'–P'**) Injection of 20 pg RNA encoding human constitutively active (CA)-Alk4 receptor, which mediates Nodal/GDF signaling during mesoderm induction, resulted in ectopic expression of *gsc* and *ta* and dorsalization of (**K'–M'**) WT embryos and (**N'–P'**) MZ*gdf3* mutants. (**Q'–V'**) Injection of 25 pg RNA encoding *Xenopus* CA-Alk2, which mediates BMP signaling for ventral patterning, had little or no effect on the dorsal expression of *gsc* or *ta* at shield stage but lead to ventralization of both WT and MZ*gdf3* embryos at 24 hpf.

DOI: https://doi.org/10.7554/eLife.28534.011

The following figure supplement is available for figure 3:

**Figure supplement 1.** (A-D) The transcriptional repressor *eve1* is expressed at the ventral and lateral margin at shield stage in WT and MZ*gdf3* embryos.

DOI: https://doi.org/10.7554/eLife.28534.012

phenotype (*Figure 3P–R,V–X*). These results indicate that Nodal signaling ligands cannot activate the nodal signaling pathway in the absence of Gdf3, and gives genetic evidence that Nodal and Gdf3 function at the same step in the signaling pathway.

Lefty proteins are negative regulators of the Nodal signaling pathway (*Bisgrove et al., 1999*; *Thisse and Thisse, 1999*). Injection of *lft1* RNA into WT embryos resulted in a loss of *gsc* expression and a reduction in dorsal expression domain of *ta* (*Figure 3Y,Z*), and lacked midline tissues including notochord, pharyngeal mesoderm and endoderm and ventral brain at 24 hpf (*Figure 3A'*). These phenotypes are indicative of Nodal deficient mutants and shared by MZ*gdf3*. Importantly, injection of *lft1* RNA into MZ*gdf3* did not exacerbate their defective patterns or phenotypes (compare *Figure 3B', C', D'* with *Figure 3D,E,F*), indicating that there no residual Nodal signaling in MZ*gdf3* embryos that can be further antagonized by exogenous Lft1.

## MZ*gdf3* mutants are responsive to non-Nodal TGFβ ligands and have an intact Nodal-downstream response pathway

Given that MZ*gdf3* embryos do not respond to overexpression of Nodal family ligand agonists and antagonists, we tested whether the receptors and other components of the downstream pathway were intact. The TGFβ ligand Activin signals through Type I and Type II Activin Receptors; subsets of which are also utilized in the propagation of Nodal signaling. To determine if loss of Gdf3 function impacted other TGFβ signaling pathways, we injected embryos with RNA encoding *Xenopus* Activin. Ectopic expression of Activin in both WT and MZ*gdf3* embryos resulted in massive ectopic expression of *gsc* and *ta* at shield stage (*Figure 3E',F',G',H',I'*). At this concentration of injected RNA, the majority of injected WT and mutant embryos died prior to 24 hpf. Surviving WT embryos exhibited slightly dorsalized phenotypes with narrower trunk and tail and kinked notochord, and a slight increase in anterior tissues (*Figure 3G'*). Exogenous expression of Activin in MZ*gdf3* embryos partially rescued the mutant phenotype at 24 hpf (*Figure 3J'*). Embryos had a thin trunk and tail with a notochord present in the anterior tail and trunk. Ventral brain tissues were also rescued as evidenced by the increase width of the head and nearly normal size eyes.

Different TGFβ ligands signal through various combinations of Type I and Type II receptors including the Type I receptors Alk2 (Acvr1) and Alk4 (Acvr1b) (zebrafish homologues *acvr1l* and *acvr1ba*, respectively). Alk4 functions as a Type I receptor for multiple TGF beta-related ligands to regulate dorsal mesoderm induction and left-right axis determination in *Xenopus* (*Chen et al., 2004*). Alk2 is important in the BMP signaling pathway (*Branford et al., 2000*), and in zebrafish (*acvr1l*) is implicated in the specification of ventral mesoderm (*Mintzer et al., 2001*). To examine whether TGFβ signaling components downstream of these receptors were functional in MZ*gdf3*, we injected RNAs encoding constitutively active (CA) constructs of *Xenopus* Alk4 and Alk2. Injection of CA-*Alk4* RNA into either WT or MZ*gdf3* embryos caused ectopic expression of both *gsc* and *ta* at shield stage (*Figure 3K', L', N', O'*). At 24hpf, injected WT embryos were partially dorsalized, and injected MZ*gdf3* embryos were partially rescued having some notochord present and largely normal anterior morphology (*Figure 3M', P'*). Exogenous expression of CA-Alk2 had the effect of slightly reducing the extent of the dorsal margin *gsc* expression domain in WT embryos while the *ta* expression pattern was not noticeably altered (*Figure 3Q', R'*). At 24 hpf, CA-Alk2 caused a severe ventralization of the embryo characterized by loss of dorsal structures such as the notochord, expanded and misshapen somites and a reduction in anterior structures including the brain and eyes (*Figure 3S'*). The absence of *gsc* expression in shield stage MZ*gdf3* was not rescued in embryos injected with CA-*Alk2*, but *ta* expression was expanded into the dorsal domain of the margin that normally lacked expression, indicating an expansion of more lateral or ventral mesoderm domains (*Figure 3T', U'*). Similar to CA-*Alk2* injected WT embryos, MZ*gdf3* embryos at 24 hpf showed a phenotype consistent with ventralization, having further reduced anterior mesoderm and neural structures including the complete loss of eyes (*Figure 3V'*). A marker for ventral mesoderm specification, the transcription factor *eve1*, is expressed in ventral mesoderm during early gastrula stages in zebrafish. Expression of *eve*1 in shield stage embryos of WT or MZ*gdf3* injected with CA-*Alk2* was expanded from the ventral domain into more dorsal regions of the margin (Figure 3—figure supplement 1). These results indicate that the downstream components of the Nodal signaling pathway are intact and capable of activation in MZ*gdf3* mutants, and suggest that Gdf3 acts at the same level in the pathway as Nodal, as an agonist for the activation of Alk4 (Acvr1ba).

## Gdf3 and Nodal must be co-expressed in dorsal midline lineages for normal development

We asked whether there was a lineage-specific requirement for the function of Gdf3 in the Nodal pathway by injecting *gdf3* or *ndr2* RNAs into a single cell at the 4–8 cell stage in WT and MZ*gdf3* embryos, with co-injection of *eGFP* RNA as a lineage tracer. We designate the injected cells as 'targeted cells,' recognizing that this retrospective lineage tracing allows analysis of the embryonic locations of the cells that expressed the injected exogenous *gdf3* or *ndr2*. Successfully injected embryos with targeted cells confined to limited sectors of the blastoderm were selected at 50% epiboly (*Figure 4A*). Morphological phenotypes of injected embryos were assessed at 24 hpf by transmitted light microscopy and with eGFP epifluorescence to detect fates of targeted cells, or were fixed at shield stage and processed for WISH to identify *gsc* expressing cells (purple) and IHC with anti-GFP (brown) to identify targeted cells.

Consistent with results from ubiquitous expression of *gdf3* (*Figure 3I*), expression of exogenous *gdf3* in subsets of cell lineages had no effect in WT embryos, irrespective of whether targeted cells were in midline-derived tissues (21 of 84 embryos) (*Figure 4B*), non-midline tissues (20 of 84) (*Figure 4C*) or more broadly distributed (43 of 84). Strikingly, MZ*gdf3* embryos were only fully rescued when *gdf3*-expressing targeted cells had contributed to midline tissues including the notochord and prechordal plate mesoderm (37 of 131 embryos) (*Figure 4D*), or in midline and non-midline tissues (62 of 131). When targeted cells were confined to non-midline tissues (32 of 131), ectopic *gdf3* expression rescued only the posterior structures of MZ*gdf3* embryos, including posterior notochord, but did not rescue anterior structures such as the ventral brain and eyes (*Figure 4E*). Results from analysis of shield stage embryos (*Figure 4J–Q*) were consistent with the results presented above for 24 hpf embryos, and those presented for 1–2 cell injections (*Figure 3G*). Ectopic *gdf3* in wild-type embryos had no effect on expression of the dorsal midline marker *gsc*, irrespective of whether targeted cells were in the dorsal midline (4 of 69 embryos), contiguous and adjacent to the midline *gsc* domain (40 of 69), or not associated with the midline (25 of 69) (*Figure 4J–M*). The ability of ectopic expression of *gdf3* to rescue *gsc* expression in presumptive shield stage MZ*gdf3* embryos was dependent on the location of the targeted cells. *gsc* expression was rescued when targeted cells were either in the midline (10 of 74 embryos, *Figure 4O*) or contiguous and adjacent to the midline (42 of 74, *Figure 4P*). In contrast, when targeted cells were confined to regions outside the dorsal midline, *gsc* expression was not rescued by ectopic *gdf3* expression (22 of 74, *Figure 4Q*). The lineage-tracing rescue data analyed at shield stage and at 24 hpf correlate well. The proportion of non-rescued embryos is about 1/3 in both cases. The proportion of embryos with targeted cells in the midline at 24 hpf is greater than the proportion of embryos with targeted cells in the dorsal midline at shield stage, most likely due to the convergence of cells toward the midline during gastrulation (*Schier and Talbot, 2005*).

In WT embryos, confinement of *ndr2*-expressing targeted cells to midline tissues (34 of 118 embryos) gave a relatively mild phenotype that included a kinked notochord (*Figure 4F*). This is in striking contrast to hyperdorsalization phenotypes with extremely disorganized tissues in WT embryos in which targeted cells contributed to non-midline tissues (32 of 118) (*Figure 4G*, *Figure 4—figure supplement 1*), or from ubiquitous *ndr2* expression (*Figure 3U*). Distribution of *ndr2* targeted cells to subsets of both midline and non-midline tissues (52 of 118) resulted in intermediate phenotypes. When analyzed at the shield stage, in WT embryos *ndr2* induces ectopic expression of *gsc* in sectors that correspond to the targeted cells, regardless of whether targeted cells are located at the dorsal midline (11 of 64 embryos, *Figure 4S*), adjacent to the midline (31 of 64, *Figure 4T*), or at a location distant from the midline (22 of 64, *Figure 4U*). In contrast to the effects of *ndr2* ectopic expression in WT embryos, targeted expression of *ndr2* in MZ*gdf3* embryos by injection at the 4–8 cell stage had no effect, that is, could not rescue, irrespective of whether targeted cells localized to prospective midline or non-midline tissues (n = 53 embryos: 11 midline, 17 non-midline, 25 broad expression) (*Figure 4H,I*). When analyzed at the shield stage, overexpression of *ndr2* in spatially restricted clones of cells in MZ*gdf3* embryos failed to induce *gsc* expression, regardless of the location of targeted cells (n = 47, *Figure 4W–Y*). Thus, in the absence of *gdf3*, regardless of lineage, *ndr2* had no apparent function; it could not induce shield-stage expression of *gsc* and could not rescue the MZ*gdf3* phenotype.

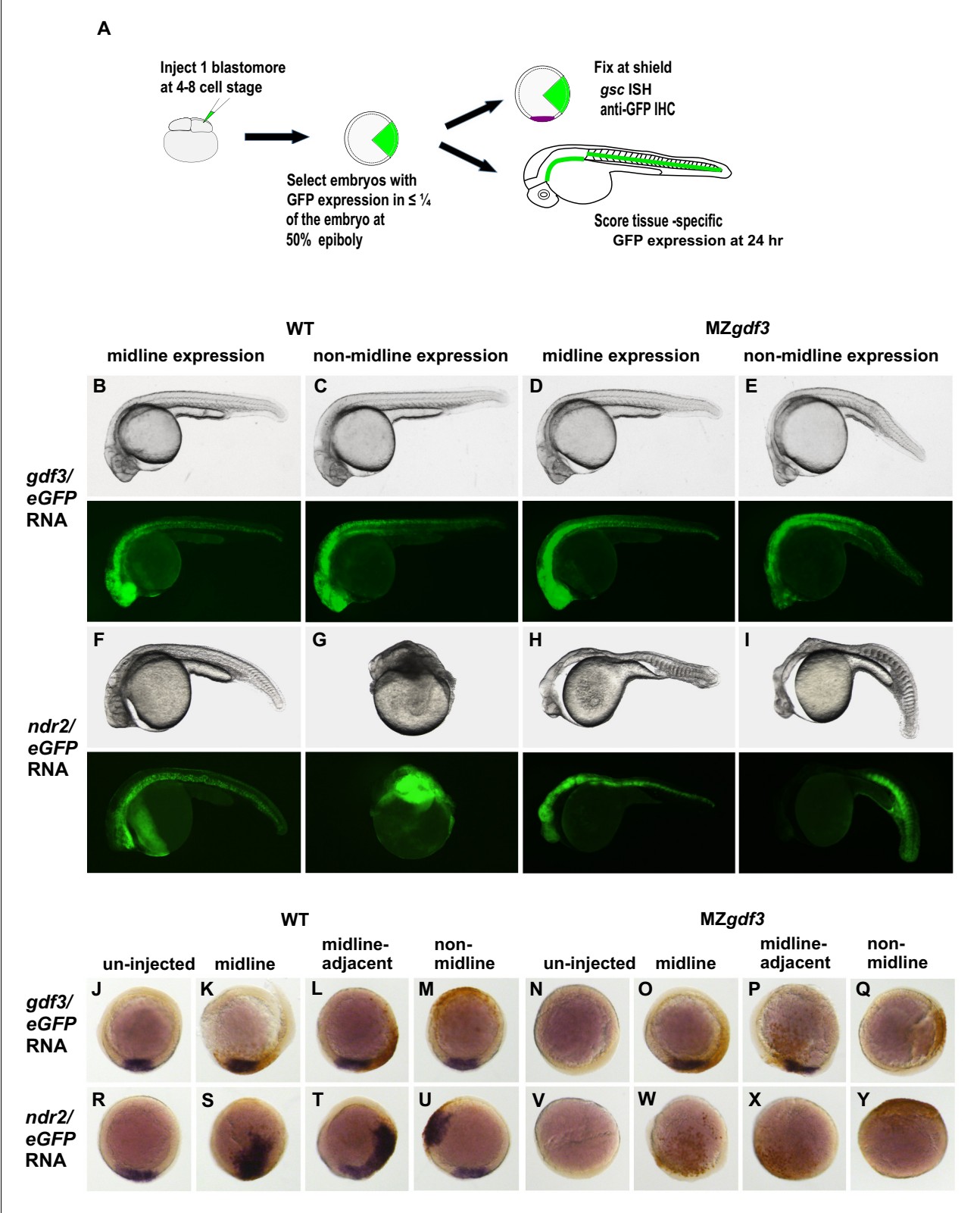

**Figure 4.** Gdf3 and Nodal must be co-expressed in lineages fated to become dorsal midline tissues. The site of ectopic Nodal/Gdf3 signaling influences the efficacy of MZ*gdf3* mutant rescue and the severity of overexpression phenotypes in WT embryos. (**A**) Experimental Approach: 4–8 cell WT and MZ*gdf3* embryos were injected with 5 pg *ndr2* RNA +25 pg *eGFP* RNA or with 50 pg *gdf3* RNA +25 pg *eGFP* RNA. At 50% epiboly embryos expressing eGFP in 25% or less of the blastoderm were selected. These embryos were grown until 24 hpf and photographed with transmitted light and
*Figure 4 continued on next page*

*Figure 4 continued*

fluorescent illumination or were grown until shield stage and processed by WISH with a *gsc* probe then by IHC with anti-GFP. (**B, C**) 24 hpf WT embryos injected with *gdf3/eGFP* RNA had normal phenotypes, regardless of whether (**B**) midline or (**C**) non-midline lineages were targeted. (**D**) MZ*gdf3* embryos expressing Gdf3/eGFP in dorsally-derived midline lineages including the notochord and polster showed complete morphological rescue of notochord, somites, brain and eyes. (**E**) MZ*gdf3* embryos with expression of Gdf3/eGFP in non-midline lineages including the skin and somites showed rescue of somites and notochord but lacked normal development of anterior neural tissues and eyes. (**F**) Expression of Ndr2/eGFP in dorsal midline lineages in WT embryos resulted in embryos that were predominantly normal but some exhibited slightly narrower head and trunk and kinked notochords. This is strikingly distinct from (**G**) expression of Ndr2/eGFP outside midline lineages, which strongly dorsalized the embryos, or from ubiquitous expression in WT embryos injected with *ndr2* RNA at the 1–2 cell stage (***Figure 3S–U***). (**H, I**) MZ*gdf3* embryos injected with *ndr2/eGFP* RNA showed no rescue of the MZ*gdf3* mutant phenotype regardless of what lineages were targeted. (**J–Y**) Shield stage embryos that were injected at the 4–8 cell stage with the indicated RNAs were processed by WISH for *gsc* and by IHC for GFP. Purple cells express *gsc* RNA; brown cells express GFP from RNA injection. Panels show embryos representative of each phenotypic class. (**J–M**) WT embryos at shield stage that were injected with *gdf3/eGFP* RNA showed no alteration in, or ectopic expression of *gsc* regardless of the location of expressing cells. (**N–Q**) MZ*gdf3* embryos injected with *gdf3/eGFP* RNA showed rescue of *gsc* expression when the expressing cells were located at (**O**), or adjacent to the presumptive dorsal shield (**P**). (**R–U**) WT embryos injected with *ndr2/eGFP* RNA showed ectopic *gsc* expression associated with the clone of expressing cells, regardless of the location of these cells within the embryo. (**V–Y**) MZ*gdf3* embryos injected with *ndr2/eGFP* RNA were unresponsive to this nodal ligand and showed no *gsc* expression. Note: Due to the lack of *gsc* expression, the location of the GFP-expressing clone of cells relative to the dorsal axis could not be reliably assigned in these embryos.

DOI: https://doi.org/10.7554/eLife.28534.014

The following figure supplement is available for figure 4:

**Figure supplement 1.** Morphological alterations caused by *ndr2* overexpression.
DOI: https://doi.org/10.7554/eLife.28534.013

The ability of ectopic *gdf* to fully rescue MZ*gdf3* embryos only when it is expressed in lineages that co-express Nodal, and the observation that MZ*gdf3* embryos are refractory to Nodal (*ndr2*) overexpression phenotypes seen in WT embryos, indicate that Gdf3 and Nodal must be co-expressed to have function, and that either ligand alone cannot function without the other.

## Discussion

Maternally supplied Gdf3 is required during early embryonic development. Zygotic *gdf3* homozygous mutant embryos have no apparent morphological phenotype, and can grow to viable and fertile adults. In contrast, embryos generated from fertilization of a *gdf3*-deficient mutant egg by a wildtype *gdf3+* sperm display the same embryonic lethal MZ*gdf3* mutant phenotype as those fertilized with sperm from a homozygous *gdf3-* mutant father. This indicates that zygotic expression of wildtype *gdf3*, confirmed by allele-specific RT-PCR, from the paternal genome after zygotic gene activation (ZGA, 4.2 hpf) (*Lee et al., 2014*), is insufficient rescue the MZ*gdf3* mutant phenotype. The inability to rescue the MZ phenotype may reflect the timing and/or levels of zygotic expression. The embryonic lethal MZ*gdf3* phenotype can however, be rescued to grow to viable fertile adults when injected with *gdf3* RNA at the 1 cell to 2 cell stages, supporting the idea that Gdf3 is essential during early embryogenesis.

Although Vg1/Gdf3 has been implicated in left-right patterning by specific cell lineage-targeting overexpression experiments in *Xenopus* (*Hyatt and Yost, 1998*) and by morpholino knockdown in zebrafish (*Peterson et al., 2013*), the zygotic *gdf3* mutants had no discernable alterations in LR patterning. There are several ways to consider the divergent zygotic mutant and morpholino LR phenotypes. First, it is possible that *gdf3* morpholinos had off-target effects. However, the LR phenotype in *gdf3* morphants could be rescued by manipulation of downstream components of the LR pathway. For example, lowering the doses of Charon, thought to be an inhibitor of the transfer of LR information from Kupffer's vesicle to LPM, rescued *gdf3* morphants (*Peterson et al., 2013*). If the agent of LR information is an obligate heterodimer of Gdf3 and Nodal, then lowering the dose of the inhibitor of this agent would be expected to rescue the partial knockdown of a component of this agent, Gdf3. Second, it is possible that in this case morpholinos provides a more acute disruption of Gdf3 function during early development, revealing roles for Gdf3 in LR patterning that are masked in the more chronic disruption of function in genetic mutants. The ability of mutants to acquire 'genetic compensation' by which some of the functions of a mutation-targeted gene are masked, has been proposed (*El-Brolsy and Stainier, 2017*) and is a topic of vigorous discussion. Third, it is possible

that maternal stores of Gdf3 persist late enough into embryogenesis to participate in LR patterning and to mask loss of zygotically encoded *gdf3*. The dramatic phenotype of the MZ*gdf3* mutants which includes a loss of lateral plate mesoderm and other tissues involved in the transfer of LR patterning signals and the inability of post-ZGA expressed paternal WT *gdf3* to rescue the MZ phenotype preclude ready assessment of this possible role of *gdf3* in LR patterning.

The loss of gene expression domains marking, in particular, neural mesodermal and endodermal tissues originating in or near the embryonic shield indicates a central role of Gdf3 in maintaining/promoting embryonic organizer function during gastrulation and is consistent with a role for Gdf3 function in the Nodal signaling pathway (*Andersson et al., 2007*; *Chen et al., 2006*; *Cheng et al., 2003*). The results of exogenous expression of the receptor-promiscuous Activin ligand and constitutively active Alk4 receptors, and their partial rescue of the MZ*gdf3* phenotype, indicates that the TGFβ response pathway is intact in MZ*gdf3*, and that Gdf3 functions upstream of the Alk4 receptor, genetically placing its function at the same level as Nodal. The fact that exogenous Nodal (Ndr1 and Ndr2) could not activate transcription of the mesodermal transcription factors *gsc* or *ta*, nor rescue the loss of midline tissues in MZ*gdf3* indicates that Gdf3 function is an essential factor in the activation of the Nodal signaling pathway, and that addition of Nodal homodimers cannot overcome the deficit of Gdf3.

Our results indicate that Gdf3 and Nodal must be co-expressed in the prospective dorsal sector of the embryo to drive embryonic patterning. Strikingly, our genetic, epistasis and lineage targeting experiments indicate that either ligand cannot function without the other ligand, so that Gdf3 is an obligate cofactor of Nodal, and Nodal is an obligate co-factor of Gdf3. Members of the Nodal and Gdf families have been shown to co-immunoprecipitate, depending on the assay and the cells in which they are co-expressed (*Fuerer et al., 2014*; *Peterson et al., 2013*), and co-expression in *Xenopus* animal cap assays induces a stronger and more distant response than either ligand alone (*Andersson et al., 2007*; *Tanaka et al., 2007*). Combining our genetic results and these previous biochemical experiments, we suggest that the functional ligand for the Nodal signaling pathway during early zebrafish development is an obligate heterodimer of Gdf3/Nodal (Gdf3/Ndr1 and/or Gdf3/Ndr2), and importantly, that homodimers of either Nodal/Nodal or Gdf3/Gdf3 are not functional in early embryonic patterning. There is precedent for this idea from studies of the role of the TGFβ family members *bmp2* and *bmp7* during zebrafish dorsoventral (DV) patterning. While homodimers of both are found in the embryo, only the heterodimer possess sufficient receptor activity to elicit the signaling required for DV patterning (*Little and Mullins, 2009*).

There is a single Nodal locus in mammals, whereas there are three Nodal-related loci in zebrafish, *ndr1*, which is maternally expressed, and *ndr2* and spaw which are zygotic (*Long et al., 2003*; *Rebagliati et al., 1998a*). The early embryonic functions of mouse Nodal, in germ layer and axial development (*Brennan et al., 2001*; *Conlon et al., 1994*; *Varlet et al., 1997*), are reflected by the functions of zebrafish paralogs Ndr1 and Ndr2 during cleavage and gastrulation (*Feldman et al., 1998*; *Rebagliati et al., 1998a*, *1998b*; *Sampath et al., 1998*). The role of murine Nodal in Left-Right patterning (*Brennan et al., 2002*; *Saijoh et al., 2003*) appears to be accomplished by Spaw and Ndr2 during somitogenesis in zebrafish (*Long et al., 2003*; *Rebagliati et al., 1998a*). In an inversion of the usual comparisons between the mammalian and duplicated zebrafish genomes, there is a single locus of the *gdf* gene family in zebrafish, *gdf3*, whereas there are two related loci in mammalian genomes, ancestral *Gdf3* and more recently derived *Gdf1*. Murine *Gdf3* mutants (*Chen et al., 2006*) phenocopy early germ layer and axial phenotypes of *Nodal* mutants while *Gdf1* mutants (*Rankin et al., 2000*; *Wall et al., 2000*) phenocopy the laterality phenotypes of node-specific *Nodal* mutants. To our knowledge, double mutants of *Gdf1* and *Gdf3* have not been reported, nor is it known whether there is a maternal contribution of either *Gdf1* or *Gdf3* that participates in murine embryonic patterning. As we have proposed in this study, and as proposed by *Peterson et al. (2013)* in zebrafish, Nodal-Gdf heterodimers are thought to be the functional entity in both germ layer and LR patterning in mice (*Andersson et al., 2006*; *Tanaka et al., 2007*). While the varying number of Gdf and Nodal family members between fish and mammals may complicate comparing genetic and biochemical outcomes between mammals and zebrafish, it provides an interesting opportunity to explore the evolution of these co-factors and their signaling pathways.

## Materials and methods

### Generation of *gdf3* mutants

TALEN nucleases targeting *gdf3* were designed and constructed in conjunction with Timothy Dahlem at the Mutation Generation and Detection Core at the University of Utah (*Figure 1—figure supplement 1*). Sequences encoding left (L) and right (R) TALENs were cloned into the pCS2 +expression vector and TALEN RNAs were synthesized using the Message Machine SP6 transcription kit (Ambion).

TALEN RNAs (25 pg each L and R) were pressure injected into 1–2 cell AB strain embryos derived from a pool of adults that had previously been sequenced to ensure genomic homogeneity ('*gdf3* clean') around the TALEN target site. Injected G0 embryos were harvested at 24 hpf and assayed by HRMA (*Parant et al., 2009*) to determine the frequency of TALEN-induced mutations in *gdf3*. Sibling embryos to those showing high frequency of *gdf3* mutations were raised to adulthood and outcrossed to '*gdf3* clean' AB adults. Clutches of F1 embryos were screened by HRMA to identify G0 parents carrying *gdf3* mutations, and sibling embryos were raised to adulthood as potential *gdf3* F1 founders, which were subsequently identified by HRMA of tailfin genomic DNA.

Wild-type and mutant zebrafish lines were maintained at the Central Zebrafish Animal Resource (CZAR) at the University of Utah. Embryos for experiments were collected from natural spawnings, cultured and staged by developmental time and morphological criteria (*Westerfield, 1995*).

### Ectopic expression assays

Capped, synthetic RNAs for injection were generated from expression constructs encoding *gdf3* (*Dohrmann et al., 1996*), *ndr*, *ndr2* (*Rebagliati et al., 1998a*, *1998b*), *lft1*, *lft2* (*Bisgrove et al., 1999*), *Xenopus activin βb* (*inhbb*) (*Thomsen et al., 1990*) and CA-*alk2* (*Hemmati-Brivanlou and Thomsen, 1995*) and human CA-*alk4* (*ACVR1B*) (*Willis et al., 1996*) using Message Machine.

Wild-type AB strain or MZ*gdf3* mutant embryos at the 1–2 or 4–8 cell stages were injected with a single or combinations of RNA. Following injection embryos were allowed to develop to desired stages, then photographed or fixed for in whole mount in situ hybridization (WISH). In all cases injection experiments were carried out on two independent clutches of embryos and a minimum of 15 embryos were examined for each experimental condition and in situ probe.

### RNA in situ localization and immunolocalization

For WISH, embryos were fixed and processed as described previously (*Bisgrove et al., 1999*). Antisense riboprobes were synthesized from linearized DNA templates using T3 or T7 polymerases and digoxigenin labeling mixes (Roche). Probes used in this study included *gsc* (*Stachel et al., 1993*), *ta* (*ntl*) (*Schulte-Merker et al., 1994*), *ndr2* (*cyc*) (*Rebagliati et al., 1998a*), *lft1*, *lft2* (*Bisgrove et al., 1999*), *tbx16* (*spt*) (*Ruvinsky et al., 1998*), *foxa2* (*axial*) (*Strähle and Jesuthasan, 1993*), *otx2* (*Li et al., 1994*), *erg2b* (*krox20*) (*Oxtoby and Jowett, 1993*). In all cases, a minimum of 15 embryos were examined for each experimental condition and in situ probe.

To label clones of cells expressing eGFP by IHC, embryos were re-fixed following WISH and incubated with peroxidase-conjugated GFP antibody diluted 1/500 (Rockland Immunochemicals Inc. #600-103-215). 3,3'-diaminobenzidine was used as color development substrate.

Following WISH and IHC, embryos were cleared in 70% glycerol/PBS and photographed with a Leica M165FC or Leica M205FA microscope Using Leica Application Suite digital acquisition software. Images were processed using Adobe Photoshop to correct color balance, exposure, brightness and contrast only.

## Acknowledgements

We thank Jan Christian for discussions of Vg1 and TGFβ superfamily. We are grateful for the exemplary collegiality of Alex Schier and Rebecca Burdine in coordinating publications.

## Additional information

### Funding

| Funder | Grant reference number | Author |
| --- | --- | --- |
| National Institutes of Health | 2UM1HL098160 | H Joseph Yost |

The funders had no role in study design, data collection and interpretation, or the decision to submit the work for publication.

### Author contributions

Brent W Bisgrove, Conceptualization, Formal analysis, Validation, Investigation, Visualization, Methodology, Writing—original draft, Writing—review and editing; Yi-Chu Su, Investigation, Visualization, Methodology, Writing—review and editing; H Joseph Yost, Conceptualization, Resources, Supervision, Funding acquisition, Methodology, Writing—original draft, Project administration, Writing—review and editing

### Author ORCIDs

Brent W Bisgrove [iD] http://orcid.org/0000-0002-0665-6401
Yi-Chu Su [iD] http://orcid.org/0000-0003-1621-6650
H Joseph Yost [iD] http://orcid.org/0000-0003-2961-5669

### Ethics

Animal experimentation: All zebrafish research was conducted in strict accordance with the recommendations in the Guide for the Care and Use of Laboratory Animals of the National Institutes of Health, and was approved by the University of Utah IACUC committee (protocol number 15-06004)

### Decision letter and Author response

Decision letter https://doi.org/10.7554/eLife.28534.017
Author response https://doi.org/10.7554/eLife.28534.018

## Additional files

### Supplementary files

• Transparent reporting form
DOI: https://doi.org/10.7554/eLife.28534.015

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
