## [Decision Letter]

Thank you for submitting your article "Maternal Gdf3 is an Obligatory Cofactor in Nodal Signaling for Embryonic Axis Formation in Zebrafish" for consideration by *eLife*. Your article has been reviewed by three peer reviewers, and the evaluation has been overseen by a Reviewing Editor and Didier Stainier as the Senior Editor. The reviewers have opted to remain anonymous.

The reviewers have discussed the reviews with one another and the Reviewing Editor has drafted this decision to help you prepare a revised submission.

Summary:

This manuscript demonstrates that, in zebrafish, maternal expression of Gdf3, the homolog of *Xenopus* Vg1, is required for mesendoderm development and subsequently for specification of the left-right axis. It is notable that Vg1, the mouse homologs Gdf1/3, and zebrafish Gdf3 have been studied before, and that it has already been shown that these factors can bind to Nodal and enhance Nodal function. It has also been shown that Gdf1,3/Vg1 is required for early development, specifically for mesoderm development. The current work expands on these findings, giving more detail on some aspects, and, importantly, uses mutants rather than antisense approaches (as previously used in *Xenopus* and in zebrafish) to reach its conclusions, giving the current work a stronger basis than the prior work. This study will likely represent a useful and important addition for the community, especially regarding future experiments addressing receptor-ligand signaling complexity, the degree to which heterodimer TGFbeta/BMP-type family signaling ligands can move in tissue, their stability, signaling potency, overlap of expression domains as a driver of signaling-instructive functions, etc. In order for the current manuscript to provide this contribution in a definitive fashion, it will be essential for the authors to develop further several elements of their results, interpretation, and presentation, as described below.

Essential revisions:

1) The authors claim that the MZ*gdf3* phenotypes are strikingly similar to the phenotypes of MZoep mutants, but further explanation of this similarity is needed to communicate the point clearly to the reader.

2) The authors do not provide the% rescue spectrum or mutant phenotype spectrum for many of the experimental manipulations. Is it true that 100% of the GDF3 RNA injected embryos are perfect recoveries of the normal phenotype, for example? This should be clarified throughout. In addition, the authors should comment on the variability in the severity of the phenotype between panels: the embryo in Figure 1, for example, is less severe than the embryo in Figure 3. lefty1 over expression in the mutant appears to produce a more severe phenotype than the mutant alone, especially with regards to ta expression in panel 3C'. Is this just variability in the phenotype that is within the phenotypic range of nodal complete loss of function or is MZ*gdf3* less severe sometimes?

3) The authors remark that Vg1 is the "only" ligand that can invert the LR axis. Isn't it the case that Nodal can also invert this axis (Ohi and Wright 2007)?

4) The authors should examine additional LR asymmetries before concluding there are no LR defects, including analysis of early markers. Analysis of heart looping at 2 dpf is insufficient evidence of normal LR asymmetry establishment.

5) This early section: ["Morpholino knockdown of *gdf3* results in.… disruption of downstream LR patterning.… zebrafish Gdf3 was proposed to transmit LR information generated by KV cilia flow to LPM (Peterson et al., 2013)"] is poorly reconciled with the later Discussion point ["Although Gdf3 has been implicated in left-right patterning.… by morpholino knockdown in zebrafish (Peterson et al., 2013), the zygotic *gdf3* mutants had no discernable alterations in LR patterning. Although possible, it seems unlikely that maternal stores of Gdf3 persist late enough into embryogenesis to participate in LR patterning.”]. Should it be more clearly raised as possible that the GDF3 morpholino results represent off-target effects? Similarly, how do the authors explain that "In contrast to previous results with morpholinos in zebrafish, zygotic mutants of *gdf3* had normal LR patterning as assessed by scoring heart looping in clutches of embryos derived from heterozygous *gdf3* in-crosses at 48 hours post-fertilization"? Does this represent inhibition of maternal contributions by the morpholinos?

Also, regarding the statement in the Discussion that "Although possible, it seems unlikely that maternal stores of Gdf3 persist late enough into embryogenesis to participate in LR patterning:" there are multiple examples of maternally loaded transcripts persisting for 2 to 4 dpf, so this isn't so unlikely. The authors should attempt to address the extent of maternal transcript persistence into zygotic development and when its implicated in LR patterning, as discussed.

6) The authors state that "At 24 hpf, MZ*gdf3* embryos lack a notochord and spinal cord and structures associated with lateral plate mesoderm including the heart, and have greatly reduced anterior neural development reflected in the narrowed brain and reduced eyes," but this is not clearly shown and could be better supported by additional images and/or histological sections.

7) The authors state: "To confirm that the MZ*gdf3* phenotype was due to a loss of wild-type *gdf3* gene product, we injected 1-2 cell MZ*gdf3* embryos with *gdf3* RNA and were able to completely rescue the *gdf3*- mutant phenotype (Figure 1)." Were a number of doses of GDF3 RNA attempted? As mentioned above, what was the spectrum of rescue in preliminary RNA dosing experiments that led to this final dose (100 pg)? If there are always 100% rescues, it would be important to add explanation of this to the paper; for example by stating that anything close to 100 pg capped synthetic RNA would work because the embryonic patterning response is controlled, essentially, by the expression domains and titers of Nodal ligand genes, such that ectopic GDF3 RNA alone cannot really have any signaling effect.

8) Regarding the RT-PCR included in Figure 1—figure supplement 2: This is not a quantitative assay and is lacking in internal controls, so the assertions about transcript levels may not be correct. If the authors did qRT PCR they could show that the transcript levels were the same. Without qPCR, the conclusion needs softening. Also, the assay does not discriminate between maternal and paternal alleles, which could very easily be done by using allele specific primers. A primer specific to the mutant deletion allele would identify the maternal allele, while a primer specific to the WT allele would identify the paternal allele. Using these primers in a quantitative assay would allow the authors to definitively address the extent of the maternal and zygotic contributions, and in a non-quantitative assay could address the question of when zygotic transcription of *gdf3* initiates. Considering the lack of a LR defect in zygotic mutants, it is possible that maternal *gdf3* persists that can heterodimerize with Nodal to specify the LR axis, so this is important to address.

9) *lft1* and *lft2* expression was examined at 90% epiboly, a timepoint long after they function in regulating mesendoderm formation, so results at this stage cannot be used to indicate that lefty is not overexpressed and the cause of the loss of mesendoderm. The authors should either test earlier stage embryos or omit this conclusion.

10) In Figure 4, the midline expression vs. non midline expression is not readily apparent. The use of the lateral view and the stage displayed do not make midline vs. non midline expression obvious. The authors should either show higher magnifications in addition to the current images or midline expression during gastrulation stages, so that it is clear whether the expression is in the midline. Also, the authors remark that the embryo in Figure 4 is mostly "undifferentiated mesoderm" without showing evidence for this.

11) The authors remark that "Expression of dvr1 RNA in a portion of the blastoderm of MZ*gdf3* was also capable of rescuing *gsc* expression in shield stage embryos (data not shown)." These data should be included in the supplement.

12) The first sentence of Discussion states "Gdf3 is required during the first few hours of embryonic development, before zygotic gene activation," but no evidence in the results is provided for a function prior to zygotic gene activation.

[Editors' note: further revisions were requested prior to acceptance, as described below.]

Thank you for resubmitting your work entitled "Maternal Gdf3 is an Obligatory Cofactor in Nodal Signaling for Embryonic Axis Formation in Zebrafish" for further consideration at *eLife*. Your revised article has been favorably evaluated by Didier Stainier (Senior Editor), a Reviewing Editor, and three reviewers.

All of the reviewers agreed that the manuscript has been improved through revision; however, there are still some remaining points that need to be addressed before acceptance, as outlined below:

1) The authors cannot state that Gdf3 function is required prior to ZGA without actually testing the timing of its FUNCTION. Just because *gdf3* is expressed maternally and its function depends on this maternal expression does not mean that it functions prior to zygotic genome activation. Many maternal transcripts persist beyond ZGA and function AFTER ZGA. This is an incorrect assumption on the authors' part and the text should be adjusted accordingly. If Gdf3 functions exclusively with Nodal in mesendoderm development and Nodal function is strictly zygotic, then that also supports a role for Gdf3 post-ZGA.

2) Related to the claim made in the second paragraph of the Discussion: It seems that the post-ZGA paternal *gdf3* expression is minimal and may be below its normal WT level based on the non-quantitative RT-PCR performed in MZ mutant embryos. This is reflected in the Results section (Figure 1—figure supplement 5, lanes 4 and 5), which qualitatively shows much lower levels of WT *gdf3* via RT-PCR at the 1000-cell stage and the 18-somite stage. The authors use this evidence (subsection “The maternal genome, not the paternal genome, dictates the phenotype of embryos”, last paragraph) to conclude that WT zygotic *gdf3* expression cannot rescue. However, it seems likely (this again is not a quantitative assay) that *gdf3* is expressed at lower levels in MZ embryos than WT embryos. This is mostly relevant to the LR function of *gdf3* and whether zygotic expression could potentially rescue the LR defect in MZ*gdf3* mutants. The authors should modify the text to reflect this point.

3) It was difficult to determine if the authors feel that Nodal-GDF heterodimers are as likely to play a role in zygotic stages in mammalian embryos (there being "probably" less maternal deposition in a mouse embryo, for example). The authors might feel that they have already covered this objectively, but it would be helpful if they could explicitly state whether they think that the evidence from mouse mutants created so far is consistent or inconsistent with this really fascinating story from zebrafish.

---

## [Author Response]

Essential revisions:1) The authors claim that the MZgdf3 phenotypes are strikingly similar to the phenotypes of MZoep mutants, but further explanation of this similarity is needed to communicate the point clearly to the reader.

Done: We have expanded the description and comparison to the nodal-pathway mutant phenotypes (paragraph 9 in Results section) and referenced Figure 1—figure supplement 3 for more detailed phenotypic analyses (also, see response to Essential revisions #6).

2) The authors do not provide the% rescue spectrum or mutant phenotype spectrum for many of the experimental manipulations. Is it true that 100% of the GDF3 RNA injected embryos are perfect recoveries of the normal phenotype, for example? This should be clarified throughout. In addition, the authors should comment on the variability in the severity of the phenotype between panels: the embryo in Figure 1, for example, is less severe than the embryo in Figure 3. lefty1 over expression in the mutant appears to produce a more severe phenotype than the mutant alone, especially with regards to ta expression in panel 3C'. Is this just variability in the phenotype that is within the phenotypic range of nodal complete loss of function or is MZgdf3 less severe sometimes?

Done: We have added a supplementary figure, Figure 1—figure supplement 4, to Results paragraph 4, that describes four different phenotypic classes of *MZgdf* rescue (also addresses Essential revisions #7), and we provide quantification of dose responses in this figure legend. We think this classification system will be useful for zebrafish field, somewhat analogous to Kao and Elinson’s *Xenopus* DAI (dorso-anterior indices). Thanks for the opportunity to provide this classification system for zebrafish. 100pg rescued 91% of mutant embryos and this is the dose we used throughout the rest of the study.

The embryos in the lefty overexpression experiments (both at gastrula and 24 hour stages) are slightly older than the control MZgdf3 embryos and the embryos in most the other panels in Figure 3 which leads to the perception that the phenotypes are more severe – this has been noted in the figure legend.

3) The authors remark that Vg1 is the "only" ligand that can invert the LR axis. Isn't it the case that Nodal can also invert this axis (Ohi and Wright 2007)?

Thanks, we were thinking of early expression in embryos. Elegant work from Chris Wright’s lab shows that later activation/expression of Xnr1 in *Xenopus* (via plasmids or transplants) can also invert the axis. This clarification and citation has been added to the Introduction paragraph 2.

4) The authors should examine additional LR asymmetries before concluding there are no LR defects, including analysis of early markers. Analysis of heart looping at 2 dpf is insufficient evidence of normal LR asymmetry establishment.

Done: We have quantitated the expression of the early L-R patterning marker *spaw* and present the data in Figure 1—figure supplement 2, and have added corresponding text to Results paragraph 2.

5) This early section: ["Morpholino knockdown of gdf3 results in.… disruption of downstream LR patterning.… zebrafish Gdf3 was proposed to transmit LR information generated by KV cilia flow to LPM (Peterson et al., 2013)"] is poorly reconciled with the later Discussion point ["Although Gdf3 has been implicated in left-right patterning.… by morpholino knockdown in zebrafish (Peterson et al., 2013), the zygotic gdf3 mutants had no discernable alterations in LR patterning. Although possible, it seems unlikely that maternal stores of Gdf3 persist late enough into embryogenesis to participate in LR patterning.”]. Should it be more clearly raised as possible that the GDF3 morpholino results represent off-target effects? Similarly, how do the authors explain that "In contrast to previous results with morpholinos in zebrafish, zygotic mutants of gdf3 had normal LR patterning as assessed by scoring heart looping in clutches of embryos derived from heterozygous gdf3 in-crosses at 48 hours post-fertilization"? Does this represent inhibition of maternal contributions by the morpholinos?Also, regarding the statement in the Discussion that "Although possible, it seems unlikely that maternal stores of Gdf3 persist late enough into embryogenesis to participate in LR patterning:" there are multiple examples of maternally loaded transcripts persisting for 2 to 4 dpf, so this isn't so unlikely. The authors should attempt to address the extent of maternal transcript persistence into zygotic development and when its implicated in LR patterning, as discussed.

Done: As suggested, we have expanded the Discussion (paragraph 2) to encompass the possibilities of morpholino off-target effects, maternal stores, and genetic compensation.

Regarding the last point of maternally loaded transcripts, we have now included analysis using WT-specific primers in the different genetic combinations and developmental stages of embryos (see also Essential revisions #8).

6) The authors state that "At 24 hpf, MZgdf3 embryos lack a notochord and spinal cord and structures associated with lateral plate mesoderm including the heart, and have greatly reduced anterior neural development reflected in the narrowed brain and reduced eyes," but this is not clearly shown and could be better supported by additional images and/or histological sections.

Done: We have added Figure 1—figure supplement 3 showing higher magnification images of structures that are disrupted or are lacking in *MZgdf3* embryos, and indicated this in Results paragraph 3.

7) The authors state: "To confirm that the MZgdf3 phenotype was due to a loss of wild-type gdf3 gene product, we injected 1-2 cell MZgdf3 embryos with gdf3 RNA and were able to completely rescue the gdf3- mutant phenotype (Figure 1)." Were a number of doses of GDF3 RNA attempted? As mentioned above, what was the spectrum of rescue in preliminary RNA dosing experiments that led to this final dose (100 pg)? If there are always 100% rescues, it would be important to add explanation of this to the paper; for example by stating that anything close to 100 pg capped synthetic RNA would work because the embryonic patterning response is controlled, essentially, by the expression domains and titers of Nodal ligand genes, such that ectopic GDF3 RNA alone cannot really have any signaling effect.

Done: dose response studies lead us to using the 100pg for full rescue, we didn’t include this in the previous version for sake of brevity. We now include this clarification in the text (Results paragraph 4) and have added a supplementary figure, Figure 1—figure supplement 4, that provides a categorization of intermediate phenotypes (that will be useful for other zebrafish studies) and quantification at different doses for *MZgdf* rescue (this also addresses Essential revisions #2)

8) Regarding the RT-PCR included in Figure 1—figure supplement 2: This is not a quantitative assay and is lacking in internal controls, so the assertions about transcript levels may not be correct. If the authors did qRT PCR they could show that the transcript levels were the same. Without qPCR, the conclusion needs softening. Also, the assay does not discriminate between maternal and paternal alleles, which could very easily be done by using allele specific primers. A primer specific to the mutant deletion allele would identify the maternal allele, while a primer specific to the WT allele would identify the paternal allele. Using these primers in a quantitative assay would allow the authors to definitively address the extent of the maternal and zygotic contributions, and in a non-quantitative assay could address the question of when zygotic transcription of gdf3 initiates. Considering the lack of a LR defect in zygotic mutants, it is possible that maternal gdf3 persists that can heterodimerize with Nodal to specify the LR axis, so this is important to address.

An internal control (B-actin) has been included as panel B in the figure (now Figure 1—figure supplement 5). We have also included panels C, and D showing analysis of the same series of samples with allele-specific primers that specifically recognize WT transcripts and have included corresponding text in the Results paragraph 6.

(Same allele-specific PCRs as in response to Comment #5).

9) lft1 and lft2 expression was examined at 90% epiboly, a timepoint long after they function in regulating mesendoderm formation, so results at this stage cannot be used to indicate that lefty is not overexpressed and the cause of the loss of mesendoderm. The authors should either test earlier stage embryos or omit this conclusion.

We have added analysis at shield stage (new supplementary figure, Figure 2—figure supplement 1) and text to Results paragraph 10, that corroborates our assertion that *lefty* is not overexpressed in *MZgdf3* at shield stage when mesendoderm is forming.

10) In Figure 4, the midline expression vs. non midline expression is not readily apparent. The use of the lateral view and the stage displayed do not make midline vs. non midline expression obvious. The authors should either show higher magnifications in addition to the current images or midline expression during gastrulation stages, so that it is clear whether the expression is in the midline. Also, the authors remark that the embryo in Figure 4 is mostly "undifferentiated mesoderm" without showing evidence for this.

We have added a series of panels (double-stained, panels J through Y) to the bottom of Figure 4 that show the correlation of expression domains of targeted RNAs and their effects on midline gene expression (*gsc*) at the dorsal shield during gastrulation. We have added text to Results paragraph 18-20 that describe this data.

We have added a supplementary figure (Figure 4—figure supplement 1) that illustrates tissue alterations in a severely dorsalized embryo and have re-worked the text in Results paragraph 20 to more accurately describe the phenotype.

The cartoon in Figure 4 and the Materials and methods section have been updated to include the IHC procedure.

11) The authors remark that "Expression of dvr1 RNA in a portion of the blastoderm of MZgdf3 was also capable of rescuing gsc expression in shield stage embryos (data not shown)." These data should be included in the supplement.

Yes, thanks, we have now included this data (Figure 4), please see response to Essential revisions #10 above.

12) The first sentence of Discussion states "Gdf3 is required during the first few hours of embryonic development, before zygotic gene activation," but no evidence in the results is provided for a function prior to zygotic gene activation.

This conclusion is based on (1) the genetics (zygotic vs. maternal-zygotic) throughout the paper and (2) the analysis of wildtype gdf3 expression brought in by the paternal genome and activated post ZGA, as confirmed by allele-specific PCR. This is discussed in the rest of the paragraph.

The genetics support for this conclusion is stated in subsequent sentence in that paragraph: “embryos generated from fertilization of a *gdf3*-deficient mutant egg by a wildtype *gdf3+* sperm display the same embryonic lethal MZ*gdf3* mutant phenotype as those fertilized with sperm from a homozygous *gdf3-* mutant father. This indicates that wildtype Gdf3 expressed from the paternal genome after zygotic gene activation (ZGA, 4.2 hpf) (Lee et al., 2014) is not sufficient to rescue the mutant phenotype.” In stark contrast to *the failure to rescue* by post-ZGA expression, the *MZgdf3* egg can be rescued by pre-ZGA expression via mRNA injection.

[Editors' note: further revisions were requested prior to acceptance, as described below.]

All of the reviewers agreed that the manuscript has been improved through revision; however, there are still some remaining points that need to be addressed before acceptance, as outlined below:1) The authors cannot state that Gdf3 function is required prior to ZGA without actually testing the timing of its FUNCTION. Just because gdf3 is expressed maternally and its function depends on this maternal expression does not mean that it functions prior to zygotic genome activation. Many maternal transcripts persist beyond ZGA and function AFTER ZGA. This is an incorrect assumption on the authors' part and the text should be adjusted accordingly. If Gdf3 functions exclusively with Nodal in mesendoderm development and Nodal function is strictly zygotic, then that also supports a role for Gdf3 post-ZGA.

The reviewer is correct, we cannot definitively state that Gdf3 functions pre-ZGA. We have modified the first sentence of the Discussion to reflect this. Nodal function in zebrafish is not strictly zygotic – this has been discussed in more detail in the final paragraph of the Discussion. (Also addresses reviewer comment 3).

2) Related to the claim made in the second paragraph of the Discussion: It seems that the post-ZGA paternal gdf3 expression is minimal and may be below its normal WT level based on the non-quantitative RT-PCR performed in MZ mutant embryos. This is reflected in the Results section (Figure 1—figure supplement 5, lanes 4 and 5), which qualitatively shows much lower levels of WT gdf3 via RT-PCR at the 1000-cell stage and the 18-somite stage. The authors use this evidence (subsection “The maternal genome, not the paternal genome, dictates the phenotype of embryos”, last paragraph) to conclude that WT zygotic gdf3 expression cannot rescue. However, it seems likely (this again is not a quantitative assay) that gdf3 is expressed at lower levels in MZ embryos than WT embryos. This is mostly relevant to the LR function of gdf3 and whether zygotic expression could potentially rescue the LR defect in MZgdf3 mutants. The authors should modify the text to reflect this point.

The reviewer raises the possibility that there is less paternal *gdf3* expression in embryos derived from MZ mutant versus WT females. This may be the case. The higher levels of expression in WT embryos versus Gdf3-/- X WT embryos during the ages where paternal expression is expected (Figure 1—figure supplement 5, lanes 2, 3 versus lanes 4, 5) may also reflect signal from maternal RNA from the WT female that has perdured into later stages, in addition to transcript from the paternal genome. It is also important to note that much of the mesoderm that normally expresses zygotic *gdf3* is missing in embryos derived from MZ mutant eggs, so it would not be surprising that less transcript is generated than in wildtype with abundant mesoderm.

We have amended the text in the Results (subsection “The maternal genome, not the paternal genome, dictates the phenotype of embryos”, last paragraph) to reflect these possibilities.

The complete loss of lateral plate mesoderm and other tissues essential for the transfer of LR patterning information precludes any assessment of rescue of LR patterning in MZgdf3 mutants.

We have added statements to the Discussion to clarify this (paragraph 2).

3) It was difficult to determine if the authors feel that Nodal-GDF heterodimers are as likely to play a role in zygotic stages in mammalian embryos (there being "probably" less maternal deposition in a mouse embryo, for example). The authors might feel that they have already covered this objectively, but it would be helpful if they could explicitly state whether they think that the evidence from mouse mutants created so far is consistent or inconsistent with this really fascinating story from zebrafish.

We have added statements to the final paragraph of the Discussion which more clearly address the specific contributions of each of the Gdf and Nodal paralogues in mice and zebrafish and referenced the relevant mutant studies.